# Quantifying GABA in Addiction: A Review of Proton Magnetic Resonance Spectroscopy Studies

**DOI:** 10.3390/brainsci12070918

**Published:** 2022-07-13

**Authors:** Claire Shyu, Sofia Chavez, Isabelle Boileau, Bernard Le Foll

**Affiliations:** 1Translational Addiction Research Laboratory, Centre for Addiction and Mental Health, Toronto, ON M5S 2S1, Canada; claire.shyu@camh.ca; 2Brain Health Imaging Centre, Centre for Addiction and Mental Health, Toronto, ON M5T 1R8, Canada; sofia.chavez@utoronto.ca (S.C.); isabelle.boileau@camh.ca (I.B.); 3Department of Pharmacology and Toxicology, University of Toronto, Toronto, ON M5S 1A8, Canada; 4Institute of Medical Sciences, University of Toronto, Toronto, ON M5S 1A8, Canada; 5Centre for Addiction and Mental Health, Campbell Family Mental Health Research Institute, Toronto, ON M5T 1R8, Canada; 6Department of Psychiatry, Division of Brain and Therapeutics, University of Toronto, Toronto, ON M5T 1R8, Canada; 7Centre for Addiction and Mental Health, Concurrent Outpatient Medical & Psychosocial Addiction Support Services, Toronto, ON M6J 1H4, Canada; 8Centre for Addiction and Mental Health, Acute Care Program, Toronto, ON M6J 1H3, Canada; 9Department of Family and Community Medicine, University of Toronto, Toronto, ON M5G 1V7, Canada; 10Waypoint Centre for Mental Health Care, Waypoint Research Institute, 500 Church Street, Penetanguishene, ON L9M 1G3, Canada

**Keywords:** GABA, γ-aminobutyric acid, magnetic resonance spectroscopy, MRS, 1H-MRS, addiction, substance use disorder, neurochemistry, neuroimaging, metabolites

## Abstract

Gamma-aminobutyric acid (GABA) signaling plays a crucial role in drug reward and the development of addiction. Historically, GABA neurochemistry in humans has been difficult to study due to methodological limitations. In recent years, proton magnetic resonance spectroscopy (^1^H-MRS, MRS) has emerged as a non-invasive imaging technique that can detect and quantify human brain metabolites in vivo. Novel sequencing and spectral editing methods have since been developed to allow for quantification of GABA. This review outlines the clinical research utilization of ^1^H-MRS in understanding GABA neurochemistry in addiction and summarizes current literature that reports GABA measurements by MRS in addiction. Research on alcohol, nicotine, cocaine, and cannabis addiction all suggest medications that modulate GABA signaling may be effective in reducing withdrawal, craving, and other addictive behaviors. Thus, we discuss how improvements in current MRS techniques and design can optimize GABA quantification in future studies and explore how monitoring changes to brain GABA could help identify risk factors, improve treatment efficacy, further characterize the nature of addiction, and provide crucial insights for future pharmacological development.

## 1. Introduction

Remarkable advances in research imaging methodologies over the last few decades have shaped and enhanced our current understanding of drug reward and addiction neurobiology. As a result, addiction, which has endured significant stigma throughout various periods in history, has started to gain its ground as a complex, chronic, relapsing brain disorder-characterized by the “compulsion to seek and take the substance, loss of control in limiting intake, and the emergence of negative affective states (e.g., mood changes, anxiety, irritability)” [1,2]. Much like chronic relapsing diseases such as hypertension, asthma, and diabetes, understanding underlying mechanisms of addiction is essential in providing effective and personalized treatment plans for affected individuals [3,4]. In recent decades, the focus of addiction research has gradually shifted towards understanding neurobiological and neurochemical changes that accompany chronic substance exposure leading to the development of addiction and addictive behaviors in humans [2,5]. Techniques such as biosensors, positron emission tomography (PET), single-photon emission computerized tomography (SPECT), and magnetic resonance spectroscopy (MRS) have been developed to study human brain neurochemistry in vivo [6,7,8].

In particular, the use of proton magnetic resonance spectroscopy (^1^H-MRS) has gained popularity given the technical advances in hardware and the availability of processing tools. MRS is currently the only non-invasive ionizing-radiation-free analytical technique that enables detection and quantification of brain neurotransmitters and metabolites in vivo. Using sequences such as 2D J-resolved MRS or an advanced MRS sequence termed MEGA-PRESS (MEscher-GArwood Point RESolved Spectroscopy), which uses targeted J-editing pulses, γ-aminobutyric acid (GABA), the primary inhibitory neurotransmitter in the central nervous system, can be quantified [9].

GABA is an integral part of normal brain functioning, and disturbances to GABA dynamics have been found to play a key role in neuropathological development, including substance dependence and addiction [1,10]. There has been growing interest in applying MRS techniques to drug discovery, including identifying new and monitoring existing medications targeting the GABAergic system for treating addiction. The GABAergic system has proven to be a promising pharmacotherapeutic target for addiction treatment and management based on current evidence from both preclinical and numerous clinical studies involving GABA modulations. In preclinical studies, positive allosteric modulators of the GABA_B_ receptor suppressed drug seeking and reinforcement behaviors in alcohol, cocaine, opioids, and nicotine self-administration paradigms [11,12,13,14]. Medications targeting the GABAergic system by modulating GABA transmission, such as the antispastic drug baclofen and the anticonvulsants gabapentin and topiramate, have all been reported to be effective in reducing craving and withdrawal symptoms in clinical trials of alcohol use disorders, as well as addiction-related endophenotypes such as impulsivity [14,15,16,17,18].

With specialized ^1^H-MRS sequences, neurochemical monitoring allows neuroscientists to further explore addiction. Uncovering GABA neurochemistry in the brain could mean opportunities for a more comprehensive range of treatment options, better patient treatment monitoring, and improved treatment retention. Evaluating GABA as a potential biomarker may also allow further understanding of individuals’ addiction risk. However, due to technical limitations related to the implementation and reliable processing of GABA measurements, the use of this method to study GABA in addiction is relatively novel. The present review aims to explore and summarize the utilization of ^1^H-MRS in humans to understand GABA neurochemistry in addiction to date. We will discuss how technical improvements can optimize MRS GABA measurements in future studies and emphasize how monitoring changes to brain GABA can better characterize the nature of addiction, improve treatment efficacy, and identify risk factors, as well as provide crucial insights for pharmacological development.

### 1.1. GABA’s Role in Addiction

In humans, GABA is the main inhibitory neurotransmitter, with a concentration of about 0.8 to 1.8 mM in a healthy human cortex for the entire metabolically active pool [19]. GABA is present in 25–50% of all synapses in the central nervous system and its concentration in the healthy human brain appears to be relatively stable over time [20,21,22]. The widespread presence and utilization of GABA also means abnormalities can result in neurotransmission and circuitry imbalance, leading to brain pathologies [1]. GABA abnormalities have been reported in preclinical and clinical MRS and post-mortem studies of seizure disorders, Parkinson’s, schizophrenia, bipolar disorders, autism spectrum disorders, and substance use disorders [19,23]. The GABAergic system also plays a crucial role in the modulation of the mesolimbic dopaminergic reward neurocircuitry, which is closely interconnected with the endogenous opioid and cannabinoid systems [1]. The balance between GABA and the main excitatory neurotransmitter glutamate, along with other key neurotransmitters such as dopamine, norepinephrine, enkephalins, dynorphin, endocannabinoids, and neuropeptides, is thus essential to maintaining normal inhibitory control, as well as proper reward, motivation, stress, and memory processing [2].

GABA in the mammalian central nervous system (CNS) is synthesized primarily from glutamate by glutamate decarboxylases GAD65 and GAD67 in the presynaptic neuron [24]. The majority of GABA synthesized is loaded into synaptic vesicles by vesicular GABA transporter (VGAT) and released into the synaptic cleft following neuron depolarization [24]. GABA is terminated when plasma membrane GABA transporters (GATs) reuptake GABA into surrounding glial cells or neurons, where GABA is then catalyzed by GABA transaminase (GABA-T) [25]. Thus, the metabolic pool quantified by MRS methods consists of GABA found intracellularly in the cytosol and vesicles within neurons and glial cells, as well as GABA released into the synapses.

GABA binds to two classes of GABA receptors: GABA_A_ and GABA_B._ GABA_A_ receptors are ionotropic, ligand-gated ion channels that predominately mediate rapid inhibitory neurotransmission throughout the CNS [13]. GABA binding results in an influx of chloride ions, which hyperpolarizes the membrane, leading to neuronal inhibition. GABA_A_ receptors are also pentameric transmembrane receptors made of five transmembrane subunits, resulting in a great diversity of GABA_A_ receptor subtypes. As a result, GABA_A_ receptors possess active and allosteric binding sites with varying degrees of binding affinity to GABAergic drugs such as benzodiazepines, barbiturates, and ethanol, which in turn may contribute to varying levels of drug sensitivity in individuals. GABA_B_ receptors are metabotropic, G-protein coupled receptors are considered to have longer acting inhibitory effects on the CNS and are the binding site for GABA agonists such as baclofen. They are thought to play a major inhibitory role in drug-related reward and reinforcement behaviors, given their wide distribution in areas of the brain involved in the mesocorticolimbic reward circuitry, such as the ventral tegmental area, nucleus accumbens, and the prefrontal cortex.

Numerous preclinical studies have reported suppression of alcohol drinking, relapse drinking, and reinforcing, rewarding, stimulating, and motivational behaviors in rats and mice following administration of GABA_B_ receptor ligands, without invoking behavioral toxicity such as sedation, hypothermia, and decreased food intake [26,27,28,29,30]. The same findings are also implicated for chronic nicotine, cocaine, and opioid administration [30,31,32,33,34]. Downregulation of GABA_B_ receptor expression, as well as a decrease in ligand binding, have also been observed in animal models of chronic drug self-administration, indicating a reduction in GABAergic activity in the brain in addictions [12]. Low GABA levels in the prefrontal cortex are found to be associated with disinhibition, poor impulse control, and decreased executive function, which are common characteristics of addiction [6]. Chronic exposure of substances that increase dopamine levels likely trigger modulation and long-term adaptations of reciprocal glutaminergic and GABAergic system within the frontal cortices, midbrain, and basal forebrain connection, contributing to the neural basis of substance dependence and addiction [35]. Indeed, preclinical and clinical studies found medications that increase GABAergic modulations to be effective in reducing drug seeking and drug reinforcement processes, as well as reducing withdrawal symptoms associated specifically with cocaine and alcohol use such as anxiety, tremors and seizures, hereby reducing the likelihood of relapse behavior [11,12,13,28,29,36,37,38,39,40,41,42].

Thus, further characterizing GABA activity and the GABAergic system presents a promising target for addiction treatments [43]. Through the use of MRS techniques, GABA can now be quantified in the brain through non-invasive means, as opposed to historical methods such as peripheral plasma measurements or cerebral spinal fluid extraction [44].

### 1.2. Proton Magnetic Resonance Spectroscopy (^1^H-MRS) in Clinical Research

Magnetic resonance spectroscopy (MRS), or nuclear magnetic resonance (NMR) spectroscopy, is a specialized technique of magnetic resonance imaging (MRI) which typically acquires signals from hydrogen protons (^1^H-MRS, also referred to as MRS in this paper) [9,10,45]. MRS is currently the only non-invasive, ionizing-radiation-free analytical technique that allows for in vivo measurements of brain biochemistry. It can detect the relative concentration of various neurotransmitters and metabolites in a localized region of the brain (MRS voxel) within minutes of scanning, and the signal acquired is visualized in terms of a spectrum, as shown in Figure 1. MRS/NMR methods have been used to characterize chemical compounds since the 1950s, but clinical application of MRS is relatively novel [23]. Since the approval of MRS for clinical use in 1995, MRS has been used to study brain tumors, cancer, stroke, seizure disorders, Parkinson’s, Alzheimer’s, anxiety, schizophrenia, bipolar disorders, and autism spectrum disorders [46,47,48,49,50,51,52]. There is also a huge potential for MRS to be utilized in clinical addiction research. A study on alcohol use disorder (AUD) patients in 1995 was the first to apply MRS to clinical addiction research–reporting measurements of choline and *N*-acetylaspartate metabolites [53]. Since then, applications of MRS in addiction research have significantly expanded. While mostly used in research settings, there is increasing evidence that it can also provide useful information for diagnostics and monitoring treatment efficacies [3,23,47,54]. Given that MRS, specifically ^1^H-MRS, uses the same standard radio-frequency coils developed for the acquisition of clinical and diagnostic MR images, MRS is more affordable and readily accessible than brain imaging tools such as PET or SPECT [55]. The magnetic field strength of the scanner measured in units of Tesla (1 Tesla = 10^4^ gauss), number of averages, timing parameters (echo time, TE and repetition time, TR), shimming methods, pulse sequences, and location and size of the MRS voxel are all crucial technical aspects to consider when designing an appropriate MRS study [56,57].

The typical neurotransmitters and metabolites quantifiable by ^1^H-MRS include *N*-acetylaspartate (NAA), creatine and phosphocreatine (tCr), choline (Cho), myo-inositol (Ins), glutamate (Glu), glutamine (Gln), and GABA (Figure 1) [21,58]. Specifically, the GABA spectrum contains three largely overlapped multiplets corresponding to the three methylene (CH_2_) groups in the molecule: the multiplet at 3.0 ppm is overlapped by a tCr peak, the one at 2.4 ppm is overlapped by Glu and Gln peaks, and the one at 1.9 ppm is overlapped by NAA, as shown in Figure 1 [9]. However, due to its low concentrations in the brain and the overlapping metabolite resonances, special MRS techniques such as J-difference editing (such as MEscher-GArwood Point RESolved Spectroscopy, MEGA-PRESS), and 2D J-resolved (J-PRESS) are required to “uncover” GABA concentrations. The MEGA-PRESS (MEscher-GArwood Point RESolved Spectroscopy) sequence uses alternating edit-ON and edit-OFF spectra to alter the GABA signature. The difference spectrum is produced by averaging and subtracting the spectra; then, the GABA peak can be quantified by specialized software such as Gannet (sample-spectrum shown in Figure 2; software available through http://www.gabamrs.com/, accessed on 30 April 2022) [9,59,60]. The 2D J-PRESS spectroscopy sequence measures GABA by quantifying its strong cross-peak coupling signature visualized on a 2D J-PRESS spectrum [61]. The 2D J-PRESS spectrum can be derived by separating the chemical shift and spin–spin coupling frequencies into two independent dimensions, as shown in Figure 14 in Rynder et al., 1995 [62]. With a second frequency dimension, the improved resolution of J-coupled metabolites allows for editing and fitting of weaker signals such as GABA. Since 2D J-PRESS is not a subtractive spectral editing technique like MEGA-PRESS, it is less affected by motion during the scan. Post-scan processing approach includes using the LCmodel or ProFit algorithm for fitting and GAMMA (software) for the generation of basis sets [63]. These techniques require specialized sequences and analytical processing pipelines, which have not typically been readily available [9,59,60]. For these reasons, GABA has been one of the least reported molecules in MRS studies, including in addiction research.

## 2. Methods

A literature search was conducted using the U.S. National Library of Medicine’s PubMed, MEDLINE, Embase, PsychINFO, and Web of Science Core Collection to identify peer reviewed original research articles involving MRS studies that report GABA in substance use disorders. Keywords used for the search include GABA, γ-aminobutyric acid, magnetic resonance spectroscopy, proton magnetic resonance spectroscopy, MR spectroscopy, MRS, 1H-MRS, addiction(s), substance use disorder(s), dependence, neurochemistry, neuroimaging, drug abuse, metabolites, alcohol, cocaine, methamphetamine, 3,4 methylenedioxymethamphetamine (MDMA), opiates, opioids, prescription opioids, marijuana, cannabis, nicotine, tobacco, alcohol use disorder, cocaine use disorder, stimulant use disorder, opioid use disorder, cannabis use disorder, and methamphetamine use disorder. Studies were included if they (1) report measurements of GABA by ^1^H-MRS, (2) contain experimental groups with chronic heavy substance users and/or individuals formally diagnosed with one or more current substance use disorder in comparison with healthy control, subgroups, or baseline before abstinence or cessation, and (3) are reported in English. Any studies involving acute exposure, light use, misuse or abuse without meeting dependence criteria, or not stating formal diagnoses were not considered. Studies involving pharmacological treatments or noting report specific MRS parameters including field strength, echo time, and repetition time were also excluded. Comorbid psychiatric conditions are common in individuals with substance use disorders and therefore were not considered an exclusion as long as findings control for the presence of the comorbidity.

## 3. GABAergic Alterations in Drug Addiction

This section reports all studies (*n* = 20) that meet the above criteria for reporting GABA in substance addictions. A summary of the findings is outlined in Table 1. Detailed methodology and demographics can be found in Table 2.

### 3.1. Alcohol

Ten studies the met criteria for reporting GABA quantification associated with AUD and abstinence in AUD individuals, though results have been contradictory. The first addiction MRS study reporting the quantification of GABA was published by Behar et al., in 1999. The cross-sectional study reported significantly lower GABA concentration in the occipital cortex of recently detoxified alcohol-dependent patients (*n* = 5) and patients with hepatic encephalopathy (*n* = 5) compared to healthy control (*n* = 10) [65]. Some participants also had recent history of nicotine and benzodiazepine use. In a cross-sectional study involving patients diagnosed with post-traumatic stress disorder (PTSD), GABA levels in the anterior cingulate cortex (ACC), but not in the parietal occipital cortex (POC) and right temporal lobe (TEMP), were higher in people with both PTSD + AUD (*n* = 10) compared to PTSD alone (*n* = 28) and to trauma exposed healthy controls (*n* = 20) [67], suggesting an increased level of GABA in PTSD individuals with alcohol dependence that is not attributable to PTSD alone. Note that participants did not have to abstain from alcohol use; therefore, results could be affected by acute alcohol exposure. In a population with comorbid bipolar disorder, AUD + bipolar individuals (*n* = 20) had lower GABA than healthy control (*n* = 19), AUD only (*n* = 20), and bipolar only (*n* = 19) populations in the dorsal ACC, but the AUD only population did not differ significantly from healthy control [68]. This same study also reported that decreased GABA was associated with higher impulsivity and craving scores [68].

In terms of longitudinal studies observing changes to GABA throughout the early alcohol abstinence period, Mason et al. [71] first reported no differences in GABA concentration between alcohol-dependent, non-tobacco smoking individuals and healthy control at 1 week and 1 month abstinence. Though taking nicotine use into consideration, GABA levels decreased over the first month of alcohol abstinence in non-smoking AUD patients (*n* = 8), but did not change in smoking AUD patients (*n* = 7); in smoking AUD patients, GABA was lower than in non-smoking patients overall and lacked abstinence-related changes [71]. This suggests that smoking may attenuate withdrawal-related changes to GABA and that changes in GABA concentration could be attributable to nicotine use during alcohol abstinence, but there is likely a complex relationship between how alcohol and nicotine play a role in modulating GABA during abstinence. Mon et al., reported no difference in GABA concentration at 1 and 4 week abstinence in both smoking and non-smoking alcohol-dependent individuals in the parietal occipital lobe (POC, *n* = 20), anterior cingulate cortex (ACC, *n* = 18) and dorsal lateral prefrontal cortex (DLPFC, *n* = 12) compared to healthy light drinkers (*n* = 16), which is consistent with Mason et al. [66]. However, GABA was not measured prior to the start of abstinence in AUD patients, so there may be abstinence-related GABA changes before or shortly after stopping alcohol (less than 7 days) that were not detected in the study. In contrast with Mason et al., smoking as a covariate did not change the lack of significance in this study [66]. These findings could be explained in part by more recent reports by Prisciandaro et al. In the dorsal anterior cingulate cortex (dACC), Prisciandaro et al., found that AUD participants had significantly lower concentrations of GABA and glutamine, but not glutamate, relative to healthy light drinkers 2.5 days following the start of abstinence, and noted that disturbances in GABA normalized after about 72 h of monitored abstinence to average healthy control level [18,69]. Specifically, GABA increased by an average of 10% between day 1 and day 3 of abstinence, but no further differences in GABA were observed between day 3 and day 7 in AUD individuals [69]. These findings suggest that neurotransmitter disturbances are likely associated with acute alcohol withdrawal, and normalization of GABA disturbances occurs relatively early, following 3–4 days of abstinence. In 2021, Wang et al., reported no difference in GABA between AUD and control on day 1 of abstinence in the ACC, but did find that GABA at day 14 of abstinence is higher than GABA at day 1 abstinence [70].

In a study involving patients with alcohol and other polysubstance dependence, they found that people with alcohol dependence + polysubstance dependence have 17–20% lower GABA than the alcohol dependence only and light drinking control group in the ACC at one month alcohol abstinence [82]. There was no significant difference between the alcohol dependence only and light drinking control groups at one month abstinence. Consistent with the abstinence timeline, this suggests that GABA changes related to chronic alcohol exposure and addiction normalize by one month of abstinence. Furthermore, note that most dependence in the polysubstance group consisted of additional cocaine use [82]. The study did not control or restrict the use of cocaine in the polysubstance dependence group.

In summary, cross-sectional findings suggest GABA may be increased or decreased in AUD before initiating alcohol abstinence, depending on the extent of alcohol exposure during the scan, which was not reported in detail by earlier findings. In longitudinal studies, GABA may be lower in AUD within 1–3 days following the start of withdrawal and abstinence compared to healthy control, and increases to likely the same level as healthy control after 3 to 14 days of abstinence. However, this change may not be detected if measuring at 7 days to 1 month of abstinence. Comorbid chronic nicotine use may impact the results by attenuating GABA changes associated with early abstinence.

### 3.2. Nicotine

Four studies on nicotine dependence reported inconsistent results but suggest there is sex-dependent modulation of GABA concentration related to tobacco smoking. In the healthy female population, it is suggested that GABA concentration fluctuates with hormonal changes associated with the female menstrual cycle, though its impact on the underlying mechanisms of neurological disorders and specifically addiction remains unclear [83,84]. Notably, GABA concentration also appears to be higher in males compared to females in healthy human brain cortices [71,85,86]. Epperson et al., first reported no change in GABA concentration in the occipital cortex of male smokers (*n* = 10) before and after 48 h of abstinence, and no significant difference compared to male healthy control (*n* = 7) [72]. Taking sex specific changes into consideration, female smokers (*n* = 6) have lower GABA compared to female healthy control (*n* = 10) [72]. Interestingly, Epperson et al., also found that female smokers lost the fluctuation of GABA levels in the occipital cortex that has been found to be associated with the menstrual cycle [83]. Consistent with healthy control findings, female smokers had lower GABA compared to male smokers. In another study, Janes et al., found no differences in the dorsal ACC between male (*n* = 11) and female smokers (*n* = 4) [74]. They reported that lower dACC GABA was associated with heightened reactivity to drug cues and more withdrawal symptoms, which suggests that GABA is related to dependence severity [74]. In the prefrontal cortex, female smokers (*n* = 15) had significantly higher levels of GABA compared to female non-smokers (*n* = 15), and female smokers had lower GABA than male smokers (*n* = 15) [73]. There was no difference in GABA between male smokers and non-smokers (*n* = 15) [73]. In a male-only nicotine and polysubstance use population, one study found no significant difference in GABA between current smoking males (*n* = 48), current smoking and polysubstance use males (*n* = 61), and healthy control (*n* = 90) [81]. While the authors reported no explanation for this result, it is in line with previous studies where they saw no changes to GABA in male smokers. In addition, it is noteworthy that the smoking-without-polysubstance-use group included individuals who also smoke cannabis, whereas the polysubstance use group did not specify which substances were involved, only that they did not meet criteria for any other substance use disorders.

In summary, evidence suggest that chronic tobacco smoking results in increased GABA in female smokers compared to non-smoking female in the occipital cortex, decreased GABA in female smokers compared to non-smoking female in the prefrontal cortex, and no changes in male smokers compared to non-smoking male for both regions. Therefore, nicotine dependence may alter GABA concentration in a sex-dependent and region-dependent manner.

### 3.3. Opioids

Only two studies to date have reported GABA measurements in chronic opioid use and opioid use disorder (OUD), with inconsistent results. A study of individuals with prescription codeine dependence still using codeine and not yet having started OUD treatment reported lower GABA and higher glutamate level in the medial PFC of OUD individuals compared to healthy controls [79]. GABA levels were negatively correlated with impulsivity and positively with cognitive performance [79]. This is consistent with the current understanding of GABA’s modulation of executive functions. In another polysubstance use population, a group of individuals with OUD receiving buprenorphine as opioid agonist therapy (*n* = 21) was assessed with an AUD group (*n* = 35) and a control group (*n* = 28), where all participants, including controls, were regular smokers [80]. No differences in GABA were found across the three groups. In addition to all groups containing smokers, all three groups also reported occasional recreational use of other substances within the last 30 days of assessments [80]. The complexity of the study population likely contributed to the lack of findings, especially given nicotine dependence has been found to alter GABA levels in a sex-dependent manner.

In summary, one recent study reported lower GABA in medial PFC in OUD patients compared to control. More studies are needed to establish the impact of opioid dependence on GABA concentration in the brain.

### 3.4. Cocaine

GABA measurements in cocaine dependence were reported in three studies. In a study with 35 patients with cocaine use disorder (CUD), including 29 also diagnosed with AUD, the authors found that the CUD and AUD group had the lowest GABA level in the left PFC, followed by the CUD-only group, followed by HC (*n* = 20) [75]. In a 2016 study, the authors reported there were no differences in GABA between the CUD group (*n* = 18) and HC (*n* = 18) in the pregenual ACC and right DLPFC, though this population included people with past or current AUD as well as cannabis and tobacco smokers, which may have affected the results [76]. A most recent study from 2021 investigating GABA in the left putamen found no differences between the cocaine-dependent (*n* = 21) and HC (*n* = 22) groups [77]. In summary, cocaine use disorder is found to be associated with lower GABA levels in the occipital cortex and no difference in the left PFC, the pregenual ACC and right DLPFC.

### 3.5. Methamphetamine

Only one study reported GABA in methamphetamine dependence. Methamphetamine-dependent users (*n* = 50) showed reduced GABA levels and altered GABA/glutamate and glutamine ratio in the left DLPFC compared to HC (*n* = 20) [78]. They also found that GABA level was negatively correlated with the duration of withdrawal, suggesting that lower GABA is associated with longer withdrawal recovery time in methamphetamine dependence [78].

### 3.6. Cannabis

One study involving adolescents with cannabis use disorder reported significantly lower GABA levels (−22%, *p* = 0.03) in the ACC in the cannabis use disorder group (*n* = 13) compared to 16 matched HC [20]. They also observed a trend of lower ACC GABA levels in people with greater cannabis use, suggesting a dose-dependent relationship for chronic cannabis exposure.

## 4. Technical Considerations of MRS in Addiction Studies

The addiction studies summarized in this review utilized one of two main GABA MRS methods: 2D J-PRESS (*n* = 6) and modified J-editing sequence (e.g., MEGA-PRESS, *n* = 13), with one exception using the short-echo semi-LASER sequence [77]. Specifically, ten studies published after 2012 employed MEGA-PRESS, whereas the three studies prior to 2012 used a similar J-editing and subtraction method [87]. Different magnetic field strengths were used, ranging from 1.5 to 7 Tesla. Lower field strengths, such as 1.5 and 2.1 Tesla, were used predominantly by earlier MRS studies. More recent studies employ higher field strengths, which have improved signal-to-noise ratios, signal detection, and better spatial resolution overall (i.e., smaller MRS voxels) [88]. The most common field strength used clinically for addiction studies is 3 Tesla, which can yield reliable and reproducible GABA measurements [21,58,89,90,91]. The regions investigated in the above addiction studies include the occipital cortex, left putamen, ACC (anterior cingulate cortex), POC (parietal-occipital cortex), DLPFC (dorsal lateral prefrontal cortex), TEMP (right temporal lobe), dACC (dorsal anterior cingulate cortex), PFC (prefrontal cortex), medial PFC, pgACC (pregenual anterior cingulate cortex), and OFC (right orbital frontal cortex). Voxel size of the regions of interest (ROI) ranges from 9 to 27 mL. A summary of the imaging details and study demographics can be found in Table 2. All studies reported used single voxel placement and with either an 8, 12, 20, or 32 channel head coil, with 8 and 32 being the most commonly used. In terms of quantitative methods, all studies prior to 2006 (*n* = 4) reported absolute concentration in mmol/Kg, with creatine as an internal reference signal. Since then, the majority of recent studies compared GABA in relative concentration as institutional unit (i.u.) (*n* = 12), with four studies from 2012 to 2021 reporting GABA in absolute concentration. Not all studies reported details on the quantification and processing methods used.

## 5. Discussion

MRS is an emerging technique that has gained popularity in clinical addiction research in the last two decades. It is evident that MRS is an invaluable tool in uncovering the neurochemistry of addiction in living human brains. However, it is also clear that MRS techniques have yet to reach their full potential and have significant room for improvements, especially for quantifying GABA [23]. Indeed, major limitations exist both within the addiction MRS studies reported to date and the MRS methodology itself.

First, most studies included here had small sample sizes that likely did not have enough statistical power to reach clinical significance. It is likely that most are preliminary studies, as MRS is just beginning to be adopted into study designs. Many studies also did not have clear clinical characterization of the populations, with a large proportion of participants reporting multiple substance use in varying amounts, leading to confounding factors that most likely contributed to inconsistencies in the results. MRS scans are also performed at various timepoints, and given the small pool of studies currently available, it is difficult to determine whether the duration and intensity of exposure and the duration and intensity of abstinence or withdrawal affected findings. A wide range of methodologies were employed, including different scanner strengths, the use of different head coils, scan parameters, different regions of interest, and their voxel size and placements. These are likely due to the lack of evidence supported guidelines regarding the use of MRS and the MEGA-PRESS sequence in humans at the time early studies were designed and conducted. Specifically, the regions of interest investigated may significantly impact GABA measurements for each substance use and likely contribute to inconsistencies in the trend of reported GABA values. Based on the summary of findings as listed in Table 1, we are also unable to determine whether there are region-dependent alterations in GABA concentration across different substance dependencies. Evidently, while most studies suggest a reduction of GABA across addiction and chronic substance exposure, there are contradictions to be investigated.

Several factors can be considered when attempting to explain the dysregulation observed in these studies. There may be changes associated with GABA synthesis, signaling, and termination that alter the availability of both GABA and its precursor, glutamate. Indeed, balanced excitatory and inhibitory activity is crucial for normal brain functioning [92]. There could be changes to GABA receptor subunit expression following long term drug exposure, leading to certain subunits being expressed more than others, possibly in a region-dependent manner, which could affect and further alter the rate of GABA receptor desensitization. On a systemic level, neuronal changes could lead to altered GABAergic neurocircuitries, leading to disturbances in stress, emotion, memory, attention, and reward processing, which then manifest as addictive behaviors [24].

Future studies focusing on better characterizing patient populations and eliminating polysubstance use as a confounding factor could significantly aid in the reliability of current findings, given that we now know that exposure to each substance, whether acute or chronic, as well as mental health conditions, all likely contribute to significant changes in GABA measurements.

Inconsistencies reported may also be a product of methodological limitations [93]. As a relatively new technique, MRS is still being optimized for clinical use [94]. While MRI scanners are considered widely available, performing MRS requires technicians who are specially trained to identify ROI and evaluate consistencies of voxel placements, which may not be immediately achievable across institutions. There are also concerns regarding the reliability and reproducibility of MRS, especially in the use of special editing sequences for measuring GABA [95]. Quantitative methodologies used in earlier studies have been highly variable, where studies utilized in-house software for post-scan processing before standardized resources were made available. A low signal-to-noise ratio achievable within a reasonable scan time is an inherent concern for MRS, which can limit accuracy in quantifying GABA, given that it occurs at a much lower concentration than other metabolites [95]. The use of smaller voxel to measure GABA in smaller brain regions can also be challenging when using common 1.5 or 3 Tesla scanners, thereby limiting the regions that can be investigated. Current MRS technique also cannot differentiate between intracellular cytosolic GABA, vesicular GABA pools, and transient synaptic GABA. Therefore, it remains difficult to elucidate the origin of the changes observed, which ultimately limits our interpretation of the findings.

Fortunately, techniques are being fine-tuned and guidelines are constantly incorporating new insights on the stability of MRS methodologies [10,21,45,46,56,57,58,59,93,94,95,96,97,98]. Resources and software for processing GABA MRS acquisitions have been made available in recent years to improve standardized post-scan processing [99,100,101]. The use of higher field strength MRI or optimization of acquisition sequence can significantly improve the application of MRS. For instance, higher field strength could enable GABA quantification in smaller brain regions such as the nucleus accumbens, where smaller voxels can benefit from the increased signal-to-noise ratio. Recent findings have also confirmed the reliability and reproducibility of measuring GABA at 3 Tesla in the insula, ACC, DLPFC, Broca’s area, cerebellar cortex, dentate nucleus, posterior cingulate cortex (PCC), and the occipital lobe, hereby supporting the use of MRS to quantify GABA at a relatively lower field strength in clinical populations [21,58,85,100,102,103,104,105,106,107]. Notably, GABA measurements are also often reported as GABA plus macromolecules, or GABA+, because the J-difference editing technique required to obtain GABA measurements also co-edits the resonance of unspecific macromolecules, resulting in an overlapping macromolecule peak at 3.0 ppm. It was found that 40–60% of the edited GABA peak may come from macromolecules [22,60]. However, preliminary studies suggest that changes to macromolecules likely do not attribute to changes in reported GABA measurements and macromolecules appears to be evenly distributed throughout the brain [22,85,108,109]. However, it is unknown whether there are inter-individual variabilities of the macromolecule contribution and distribution, specifically in substance use disorders [110]. This remains to be investigated. Overall, we can expect to see considerable advances in clinical MRS with improved methods of macromolecule suppression, partial volume correction, spectral editing, and better hardware and analysis techniques. While the continuous implementation of improved methodologies may imply high heterogeneity and variation in outcome measures, it is nonetheless useful both for informing future clinical addiction studies and future MRS guidelines. It is therefore imperative for future MRS studies to consider reporting detailed methodology, at least to the extent suggested by current guidelines, in order to aid in characterizing and optimizing future MRS applications [98].

In recent years, an increasing number of clinical trials have also employed MRS to help understand GABA’s relationship with treatment efficacy of currently approved or off-label treatments for addictions [47]. This approach could be used to monitor GABA changes in relation to dose-dependent treatment response and to help identify potential patient subgroups in clinical trials. A recent cognitive behavioral therapy (CBT) treatment study reported that adolescents with internet and phone addictions had higher GABA levels in the ACC compared to healthy adolescents prior to starting CBT treatment, and that GABA levels became comparable to healthy adolescents following 9 weeks of CBT [111]. This is opposite from the trend typically seen in other substance use disorders, and it is possible that there are differences in the mechanisms involved in behavioral addiction. Transcranial direct current stimulation has recently been tested in individuals with gambling disorders (*n* = 16) [112]. In a sham-controlled, crossover, randomized study, one study found elevated GABA levels in the DLPFC, but not the striatum during active stimulation, suggesting that non-invasive techniques may be of therapeutic interest, given that elevated GABA is generally associated with less impulsivity and better executive function [112]. Indeed, GABA increase has already been associated with good treatment efficacy in depression and epilepsy [113,114]. An AUD treatment trial revealed that changes to GABA levels are closely related to days of abstinence, and that those with more relapses respond more significantly to gabapentin administration, which is a GABA analog medication known to increase brain GABA levels. This study found that gabapentin was associated with greater increases in GABA levels for those with difficulty staying abstinent. Current evidence suggests that gabapentin is moderately effective in reducing symptoms of alcohol withdrawal and craving, though it is not effective in treating AUD [15]. Given this, monitoring GABA levels during addiction treatment could shed light on treatment variabilities [16]. Baclofen, a GABA_B_ receptor agonist, has also been a drug of interest for the treatment of AUD [38,40], though evidence is inconclusive. Systematic reviews pointed to the importance of identifying potential moderators and mediators of baclofen’s effects on alcohol use to help explain the variations in responses to baclofen in AUD populations. Monitoring brain GABA in different AUD populations might in fact help elucidate these inconsistencies. A recent study reported that lower GABA is associated with the severity of alcoholic liver disease in AUD patients [115]. Clinical trials also suggest baclofen may be more effective in promoting alcohol abstinence specifically in AUD patients with alcoholic liver disease compared to AUD individuals with no alcoholic liver disease [39,41,42]. This suggests that there may be a close relationship between GABA concentration and the treatment efficacy of baclofen in the subgroup of AUD patients with alcoholic liver disease, regardless of baclofen’s efficacy for treating AUD in general. Overall, recent studies demonstrate that MRS can measure changes in GABA levels following interventions, and the difference in the degree of neurochemical changes may help explain why individuals respond to treatments differently.

## 6. Conclusions

While the use of MRS in addiction research is still in its infancy, it is promising, invaluable, and the only non-invasive technique that can quantify metabolite concentrations in vivo. It is also more widely accessible and more cost effective to employ than other imaging techniques. MRS research on the mechanisms underlying reward and addiction could help us understand why certain individuals are at lower risk of substance dependence or addiction, and why certain individuals with addiction can recover while others experience multiple relapses. Huge gaps remain in characterizing GABA in different substance use disorders and more studies are needed to understand GABA’s relationship with addiction. Future addiction imaging research should consider incorporating GABA quantification into study designs, especially in protocols that already utilize MR imaging. For AUD, longitudinal studies that closely monitor before and during abstinence could help resolve the contradictory evidence reported to date. Taking the latest GABA reporting consensus into consideration will also significantly improve the reliability and reproducibility of MRS addiction studies and promote a more standardized approach moving forward. Different regions of interest can also be explored, including the insula, which has been found to play a major role in drug craving and mediating other addiction phenotypes, such as executive functioning [116,117,118,119]. Ultimately, by understanding how medications and addictions change the GABAergic system and how changes modulate behaviors, we can develop a clearer picture of the underlying neuropathology of addiction and contribute to the development of novel addiction therapies.

## Figures and Tables

**Figure 1 brainsci-12-00918-f001:**
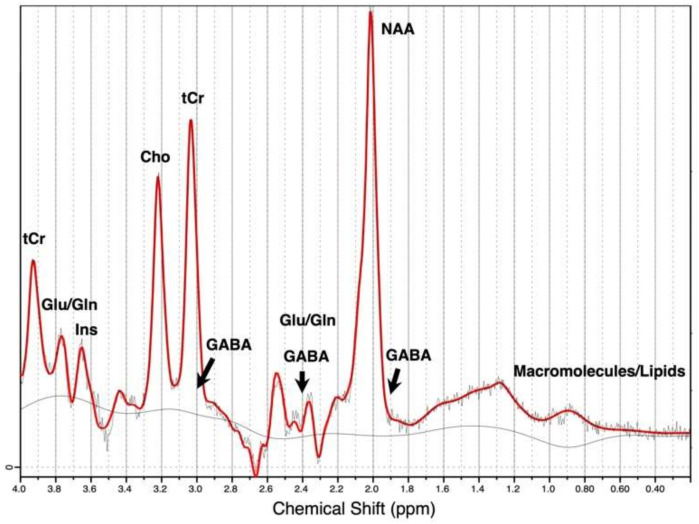
Sample in vivo ^1^H-MRS spectrum output from LCModel (software available through http://s-provencher.com/lcmodel.shtml, accessed on 30 April 2022) for data acquired at 3 Tesla [64]. Brain metabolites commonly reported by ^1^H-MRS are quantified and labelled at their corresponding chemical shift. *N*-acetylaspartate (NAA), creatine and phosphocreatine (tCr), choline (Cho), myo-inositol (Ins), glutamate (Glu), glutamine (Gln).

**Figure 2 brainsci-12-00918-f002:**
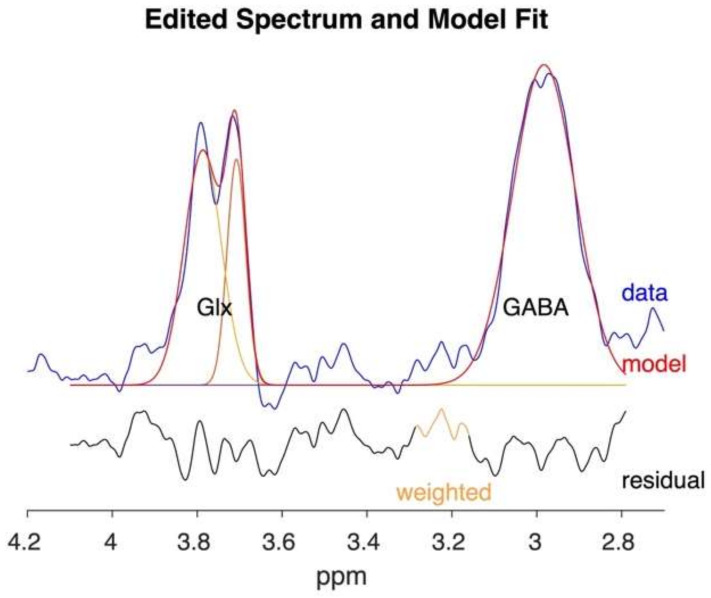
Sample edited spectrum and model fit output from Gannet (v3.1, software available through http://www.gabamrs.com/, accessed on 30 April 2022) showing modelling of GABA signal from the MEGA-PRESS sequence. As preprocessing and fitting tools for MRS edited spectra such as Gannet become more readily available, more institutions can use the newer MEGA-PRESS sequence to quantify GABA systematically [45].

**Table 1 brainsci-12-00918-t001:** Summary of findings from MRS addiction studies reporting GABA measurements.

	ACC	dACC	pgACC	Occipital Cortex	POC	TEMP	DLPFC	PFC	OFC	Left Putamen
Alcohol	-/↑ */↓	-/↓		↓	- *	-	- *		-	
Nicotine		-		↓ (Female only)				↑		
Opioids	-				-		-	↓	-	
Cocaine			-				-	↓		-
Methamphetamine							↓			
Cannabis	↓									
Polysubstance	-/↓	-			-		-		-	

[-] represents no change observed in the region of interest. Each symbol represents a finding reported from a study. ↓ indicates lower GABA in drug dependence compared to control, ↑ indicates higher GABA in drug dependence compared to control. Shaded areas indicate no reported findings to date. * Longitudinal studies comparing late alcohol abstinence data with early abstinence. Abbreviations: ACC (anterior cingulate cortex), POC (parietal-occipital cortex), DLPFC (dorsal lateral prefrontal cortex), TEMP (right temporal lobe), dACC (dorsal anterior cingulate cortex), PFC (prefrontal cortex), pgACC (pregenual anterior cingulate cortex), OFC (right orbital frontal cortex).

**Table 2 brainsci-12-00918-t002:** Published ^1^H-MRS research in individuals with substance dependence. Imaging methods utilized scanners with field strengths between 1.5–7 Tesla. All studies used single voxel placement method and used 8, 12, 22, or 32 channel head coils. Most studies employed MEGA-PRESS or J-PRESS editing sequences to quantify GABA.

Substance	Study	Subjects	Sex (Male/Female)	Mean Age (SD)	Imaging Details	Region Volume (Voxel Dimensions)	Results on GABA	Clinical Correlations
Alcohol	Behar et al., 1999 [65]	5 AUD 5 hepatic encephalopathy 10 HC	NA	46 (11) 40 (6) 35 (7)	2.1 T J-edited TR/TE = 3390 ms/68 ms	Occipital cortex 13.5 mL	↓ GABA in AUD ↓ GABA in patients with hepatic encephalopathy	NA
Alcohol	Mon et al., 2012 [66]	20 alcohol-dependent (starting abstinence) 16 light drinking control	17/3 14/2	54 (9) 49 (10)	4 T 8 channel coil MEGA-PRESS TR/TE = 2000 ms/71 ms Scan time: 12.5 min	ACC 17.5 mL (35 × 25 × 20 mm^3^) POC 16 mL (20 × 40 × 20 mm^3^) Right DLPFC 16 mL (40 × 20 × 20 mm^3^)	No difference between 1- and 4-week abstinence in both smoking and non-smoking AUD compared to control	NA
Alcohol	Pennington et al., 2014 [67]	10 PTSD + AUD 28 PTSD 20 trauma exposed control	10/0 28/0 20/0	52 (14) 35 (11) 36 (12)	4 T 8 channel coil MEGA-PRESS TR/TE = 2000 ms/71 ms Scan time: 12.5 min	ACC 17.5 mL (35 × 25 × 20 mm^3^) POC 16 mL (20 × 40 × 20 mm^3^) TEMP 16 mL (20 × 40 × 20 mm^3^)	↑ GABA in ACC for PTSD + AUD compared to both PTSD and Control no difference in GABA in temporal lobe and POC	↑ GABA in temporal lobe and ACC correlated with better neurocognition
Alcohol	Prisciandaro et al., 2017 [68]	20 bipolar disorder (BD) +AUD 19 BD only 20 AUD only 19 HC	13/7 11/9 12/7 11/8	37 (13) 36 (11) 43 (12) 38 (11)	3 T 32 channel coil 2D J-PRESS TR/TE = 2400 ms/31–229 ms Scan time: 13:28 min	dACC 18.8 mL (25 × 25 × 30 mm^3^)	↓ GABA in BD + AUD compared to HC, AUD only and BD only groups No difference between AUD only group and HC	↓ GABA in dACC correlated with higher impulsivity
Alcohol	Prisciandaro et al., 2019 [18]	20 AUD (starting abstinence) 20 light drinking control	15/5 11/9	27 (6) 24 (3)	3 T 2D J-PRESS TR/TE = 2400 ms/31–229 ms; Scan time: 13:28 min	dACC 18.8 mL (25 × 25 × 30 mm^3^)	↓ GABA in AUD relative to light drinking control 2.5 days after last drink	NA
Alcohol	Prisciandaro et al., 2020 [69]	23 AUD (starting abstinence) 12 light drinking control	18/5 NA	27 (6) 24 (3)	3 T 2D J-PRESS TR/TE = 2400 ms/31–229 ms Scan time: 13:28 min	dACC 18.8 mL (25 × 25 × 30 mm^3^)	↓ GABA in AUD relative to light drinking control <1 day after last drink Increased GABA (10%) on day 3 of abstinence compared to day 1 in AUD No change in AUD GABA between day 3 and day 7 of abstinence	NA
Alcohol	Wang et al., 2021 [70]	20 AUD (starting abstinence) 22 HC	15/5 18/4	46 (10) 46 (12)	3 T 32 channel coil MEGA-PRESS TR/TE = 3000 ms/68 ms	ACC 24 mL (20 × 30 × 40 mm^3^)	no difference between AUD and control on day 1 of abstinence Controlling for benzodiazepine use, increased GABA on day 14 compared to day 1 of abstinence in AUD	NA
Nicotine+ alcohol	Mason et al., 2006 [71]	12 AUD (5 non-smokers, 7 smokers, starting abstinence) 8 HC	12/0 8/0	39 (8) 39 (9)	2.1 T 7-cm surface coil J-edited TR/TE = 2000 ms/68 ms	Occipital cortex 13.5 mL (30 × 30 × 15 mm^3^) 20.5 mL (35 × 15 × 39 mm^3^)	no difference between AUD and control Non-smoking AUD had ↑GABA compared to smoking AUD at 1 week abstinence, then decreased to same level as smoking AUD at 1 month abstinence No change in GABA for smoking AUD at 1 week and 1 month abstinence	NA
Nicotine	Epperson et al., 2005 [72]	16 smokers 20 HC	10/6 7/13	37 (18)/35 (13) 34 (6)/31 (6)	2.1 T 7-cm surface coil J-edited TR/TE = 3390 ms/68 ms	Occipital cortex 13.5 mL (15 × 30 × 30 mm^3^)	↓ GABA in female smokers compared to male smokers ↓ GABA in female smokers compared to female HC in follicular phase, and lost GABA changes associated with menstrual cycle No change in male before and after 48-h abstinence No difference in male smoking and male HC	NA
Nicotine	Baggaa et al., 2018 [73]	30 smokers 30 HC	15/15 15/15	25 (3)/25 (3) 23 (3)/26 (4)	3 T 32 channel coil MEGA-PRESS TR/TE = 2000 ms/68 ms 256 averages	PFC 27 mL (30 × 30 × 30 mm^3^)	↑ GABA in smokers compared to HC ↓ GABA in female smokers compared to male smokers ↑ GABA in female smokers compared to female non-smokers No difference in male smokers and non-smokers	NA
Nicotine	Janes et al., 2013 [74]	15 smokers	7/8	26 (5)	3 T 32 channel coil 2D J-PRESS TR/TE = 2250 ms/30–350 ms Scan time: 12 min	dACC 12 mL (20 × 20 × 30 mm^3^)	No differences between male and female No difference between female in luteal and follicular phase	↓ GABA correlated with higher reactivity to smoking cues ↓ GABA correlated with more withdrawal symptoms
Cocaine	Ke et al., 2004 [75]	35 cocaine use disorder (12 with concurrent AUD) 20 HC	26/9 7/13	43 (7) 39 (8)	1.5 T 2D J-PRESS TR/TE = 2320 ms/48 ms	Left PFC 18.75 mL	↓ GABA in cocaine-dependent compared to HC ↓ GABA in cocaine-dependent +AUD compared to HC	NA
Cocaine	Hulka et al., 2016 [76]	18 CUD 18 HC	18/0 18/0	36 (8) 36 (8)	3 T 2D J-PRESS TR/TE = 1600 ms/26–224 ms Scan time: 22 min	pgACC 9 mL (25 × 18 × 20 mm^3^) Right DLPFC 9 mL (25 × 18 × 20 mm^3^)	No GABA difference between CUD and HC	↑ GABA associated with longer alcohol abstinence duration in pgACC ↓ GABA associated with longer alcohol abstinence duration in rDLPFC
Cocaine	Ersche et al., 2021 [77]	21 CUD 22 HC	21/0 22/0	42 (11) 38 (11)	7 T 32 channel coil Short-echo semi-LASER sequence TR/TE = 5000 ms/26 ms	Left Putamen 9 mL (16 × 16 × 35 mm^3^)	No difference in GABA	NA
Methamphetamine	Su et al., 2020 [78]	50 MUD 20 HC	NA	32 (7) 29 (6)	3 T 12 channel coil MEGA-PRESS TR/TE = 1800 ms/68 ms Scan time: 10:20 min	Left DLPFC 8 mL (20 × 20 × 20 mm^3^)	↓ GABA in MUD compared to HC	↓ GABA correlated with longer withdrawal duration
Cannabis	Prescot et al., 2013 [20]	13 adolescents with cannabis use disorder 16 HC	11/2 7/9	18 (1) 16 (2)	3T 12 channel coil MEGA-PRESS	ACC 15 mL (30 × 25 × 20 mm^3^)	↓ GABA in adolescent with cannabis use disorder	NA
Rx opioids	Li et al., 2020 [79]	31 OUD 32 HC	29/2 25/7	25 (4) 24 (4)	3 T 20 channel coil MEGA-PRESS TR/TE = 2000 ms/68 ms Scan time: 8:40 min	Medial PFC 12 mL (23 × 23 × 23 mm^3^)	↓ GABA in OUD compared to HC	↓ GABA correlated with higher impulsivity scores ↓ GABA correlated with lower cognitive function scores (Montreal cognitive assessment, MoCA)
opioids, alcohol	Murray et al., 2016 [80]	21 OUD on buprenorphine 35 AUD (abstinent for 3 weeks) 28 HC	13/8 29/6 24/4	41 (12) 47 (9) 44 (9)	4 T 8 channel coil MEGA-PRESS TR/TE = 2000 ms/71 ms Scan time: 2.5 min	ACC 17.5 mL (35 × 25 × 20 mm^3^) Right DLPFC 16 mL (20 × 40 × 20 mm^3^) OFC 8 mL (40 × 20 × 10 mm^3^) POC 16 mL (40 × 20 × 20 mm^3^)	No difference in GABA across the 3 groups	NA
Polysubstance	Schulte et al., 2017 [81]	30 smokers 38 smoking p olysubstance users 61 HC	48/0 61/0 90/0	36 (11) 33 (7) 32 (10)	3 T 32 channel coil MEGA-PRESS TR/TE = 2000 ms/73 ms	dACC 10.5 mL (35 × 20 × 15 mm^3^)	No difference in GABA across the 3 groups	NA
Polysubstance	Abe et al., 2013 [82]	40 AUD 28 AUD + other substance use disorder 16 light drinking control	37/3 26/2 15/1	52 (9) 45 (10) 49 (10)	4T 8 channel coil MEGA-PRESS	ACC 17.5 mL (35 × 25 × 20 mm^3^) POC 16 mL (20 × 40 × 20 mm^3^) right DLPFC 16 mL (40 × 20 × 20 mm^3^)	↓ GABA in ACC of AUD + polysubstance use disorder compared to AUD and control at one month abstinence	↓ GABA in DLPFC correlated with higher cocaine consumption

↓ indicates lower GABA in drug dependence compared to control or as described, ↑ indicates higher GABA in drug dependence compared to control or as described. Abbreviations: HC (healthy control), ACC (anterior cingulate cortex), POC (parietal-occipital cortex), DLPFC (dorsal lateral prefrontal cortex), TEMP (right temporal lobe), dACC (dorsal anterior cingulate cortex), PFC (prefrontal cortex), pgACC (pregenual anterior cingulate cortex), OFC (right orbital frontal cortex).

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
