# Peer review of "Quantifying GABA in Addiction: A Review of Proton Magnetic Resonance Spectroscopy Studies"

_brainsci, 2022, doi:10.3390/brainsci12070918_

Round 1
Reviewer 1 Report
The authors evaluate the role of the primary inhibitory neurotransmitter in the central nervous system gamma-aminobutyric acid (GABA) in normal brain functioning and how alterations in the GABA levels play a key role in neuropathological development, including substance dependence and addiction.
The authors review more than 100 studies in which GABA level was measure by Proton Magnetic Resonance Spectroscopy (1H-MRS). This technique is the only non-invasive, ionizing-radiation-free technique that enables detection and quantification of brain neurotransmitters and metabolites in-vivo and has gained popularity due to technical advances in hardware and the availability of processing tools.
The authors found increased or decreased GABA level in alcohol use disorders before initiating alcohol abstinence, depending on the extent of alcohol exposure during scan, alteration in GABA concentration in a sex dependent and regional dependent manner of nicotine users; lower GABA in medial prefrontal cortex in opioid use disorder patient; lower GABA levels in the occipital cortex of cocaine use patients; lower GABA levels in the anterior cingulate cortex of people with greater cannabis use.
Contradictory and inconsistent results are the major limitation to this review. Most studies included here obtained small sample sizes that likely did not have enough statistical power to reach clinical significance. Moreover, the authors mention the sequences used for acquiring the spectra but do not include the quantitative protocol adopted (the reference signal for absolute quantification, short echo time to avoid T2 influences, long repetition time to avoid T1 weighted, etc.). I think that a sentence regarding this matter in the point 4 (Technical Considerations of MRS in Addiction Studies) would improve the review.
The findings of the present work are interesting and innovative, achieved with MRS sequences commonly adopted in MR centers. The paper is well written in a synthetic way. The title describes the rationale and the methodology. The introduction focusses on the problem. Methodological details have been provided. The results are clearly described. The number of tables is appropriate. The discussion highlights the major results and the limitations of the papers included. References are pertinent and upgraded.
To improve the manuscript, I suggest to include the considerations about of the adoption of an absolute quantification method.
Author Response
Dear Reviewer,
Please see attachment for our response letter.
Thank you for your consideration.

Reviewer 2 Report
Well written review manuscript. Timely and an important area to invest. Authors have done an excellent job compiling relevant literature and discussing pitfalls of several studies and hence important cautionary notes in the interpretation of data and conclusions. Some minor points needs attention:
1) Line 113, please include "..... vesicles in neurone and glia"
2) Line 232, omit HC as this is not in table 1
3) Worth discussing voxel size in lower field MRS and how this interferes with metabolite measurements in small brain areas such as the nucleus accumbens.
Author Response
Dear Reviewer,
Please see the attachment for our response letter.
Thank you for your consideration.

Reviewer 3 Report
The review manuscript submitted to Brain Sciences by Shyu et al. concerns the role of GABA as a central player in brain metabolism, its role in metabolic linked diseases and as a target for treatment of addiction with focus on the use of 1H-MRS to characterize GABA´s role and action in that context.
The manuscript gives a good introduction to the subject with relevant old and new literature in the field. It is well structured and easy to read. My main criticism concerns the review of the selected 1H-MRS studies where many of the included studies are old and thus have no quantitative measurements of GABA. It would be helpful to show in addition to Figure 1 and 2 a state-of-the-art experiment using the two j-res methods and showing which type of quantitative data it is possible to derive. Such Figures and numbers could then be supporting an added column in Table 2 that would allow the reader to overview, which studies that are merely developing methods and which ones that add clinical data to be included when discussing the value of using 1H-MRS as a clinical method.
A minor critic is the appearance of Figure 1 and 2. It would help the reader to enhance the quality of the Figures (especially the fond size on the x-axis in Figure 1).
Author Response
Dear Reviewer,
Please see attached letter for our response to the reviewers' comments.
Thank you for your consideration.
